# Parathyroid Hormone Induces Human Valvular Endothelial Cells Dysfunction That Impacts the Osteogenic Phenotype of Valvular Interstitial Cells

**DOI:** 10.3390/ijms23073776

**Published:** 2022-03-29

**Authors:** Mihaela Vadana, Sergiu Cecoltan, Letitia Ciortan, Razvan D. Macarie, Andreea C. Mihaila, Monica M. Tucureanu, Ana-Maria Gan, Maya Simionescu, Ileana Manduteanu, Ionel Droc, Elena Butoi

**Affiliations:** 1Biopathology and Therapy of Inflammation, Institute of Cellular Biology and Pathology “Nicolae Simionescu”, 050568 Bucharest, Romania; mihaela.vadana@icbp.ro (M.V.); sergiu.cecoltan@icbp.ro (S.C.); letitia.ciortan@icbp.ro (L.C.); razvan.macarie@icbp.ro (R.D.M.); andreea.mihaila@icbp.ro (A.C.M.); monica.pirvulescu@icbp.ro (M.M.T.); anca.gan@icbp.ro (A.-M.G.); maya.simionescu@icbp.ro (M.S.); ileana.manduteanu@icbp.ro (I.M.); 2Cardiovascular Surgery Department, Central Military Hospital, 010825 Bucharest, Romania; ionel.droc@gmail.com

**Keywords:** human valvular endothelial cells, PTH, endothelial dysfunction, osteogenic molecules, valvular interstitial cells

## Abstract

Parathyroid hormone (PTH) is a key regulator of calcium, phosphate and vitamin D metabolism. Although it has been reported that aortic valve calcification was positively associated with PTH, the pathophysiological mechanisms and the direct effects of PTH on human valvular cells remain unclear. Here we investigated if PTH induces human valvular endothelial cells (VEC) dysfunction that in turn could impact the switch of valvular interstitial cells (VIC) to an osteoblastic phenotype. Human VEC exposed to PTH were analyzed by qPCR, western blot, Seahorse, ELISA and immunofluorescence. Our results showed that exposure of VEC to PTH affects VEC metabolism and functions, modifications that were accompanied by the activation of p38MAPK and ERK1/2 signaling pathways and by an increased expression of osteogenic molecules (BMP-2, BSP, osteocalcin and Runx2). The impact of dysfunctional VEC on VIC was investigated by exposure of VIC to VEC secretome, and the results showed that VIC upregulate molecules associated with osteogenesis (BMP-2/4, osteocalcin and TGF-β1) and downregulate collagen I and III. In summary, our data show that PTH induces VEC dysfunction, which further stimulates VIC to differentiate into a pro-osteogenic pathological phenotype related to the calcification process. These findings shed light on the mechanisms by which PTH participates in valve calcification pathology and suggests that PTH and the treatment of hyperparathyroidism represent a therapeutic strategy to reduce valvular calcification.

## 1. Introduction

Vascular calcification—regarded lately as a highly regulated, cell-mediated process—is associated with increased cardiovascular mortality, particularly in patients with chronic kidney disease (CKD) [1]. Parathyroid hormone (PTH), secreted by the parathyroid gland, is a crucial regulator of calcium, phosphate, and vitamin D metabolism. Secretion of PTH is triggered by low serum calcium and results in raised levels of calcium through its release from the bones or reduction of renal excretion. Moreover, released PTH inhibits the reabsorption of phosphate in the renal proximal tubule, thus generating hypercalcemia and hypophosphatemia [2].

PTH is one of the major players in CKD and mineral bone disorder [3]. In addition, it has also been reported to have effects on the cardiovascular system, including the heart and blood vessels [4]. A meta-analysis concerning the association between serum PTH levels and cardiovascular diseases (CVD) found that a high level of PTH is associated with an enhanced risk for overall CVD events [4].

PTH receptors are expressed throughout cells of the cardiovascular system, including cardiomyocytes, endothelial and smooth muscle cells [5,6]. Over the last few years, clinical studies found that PTH is associated with cardiovascular events and mortality irrespective of whether the patients have CKD or not [7]. Also, PTH was found to participate in the development of vascular calcification, a process that is accelerated in patients with CKD [8]. Vascular calcification has been frequently documented in primary hyperparathyroidism, although its pathophysiology is still a matter of debate.

Increased levels of PTH were found to correlate with the progression of the aortic valve calcification process and with extended calcification area of the aortic valve [9,10], but the precise role of this hormone in calcific aortic valve disease (CAVD) is unclear. Since ageing and atherosclerosis—factors associated with valvular calcification—are present in only 50% of patients with severe aortic stenosis [11], discovering if aortic valve calcification is associated with PTH levels may provide important insights into the pathogenesis of CAVD. Thus, other factors must play a role in the development of aortic calcification, and a higher level of PTH may be one of them.

Increasing evidence demonstrates that similar to vascular calcification, aortic valve calcification is an active process resembling osteogenesis [12] that involves osteoblast transformation of valvular cells [13].

The trans-differentiation of valvular interstitial cells (VIC) into osteoblast-like cells is thought to be the essential pathophysiological factor in valvular calcification [14], but the role of valvular endothelial cells (VEC), which are crucial in maintaining valvular integrity and homeostasis, is less studied. Dysfunctional VEC are a major contributor to valvular disease initiation and progression, and under certain conditions, they also participate in aortic valve calcification via endothelial-mesenchymal transition (EndMT) [15], cytokine secretion or regulation of angiogenesis [16]. As in atherosclerosis, the integrity of VEC is crucial in the maintenance of valve physiology and function. In injured aortic valve leaflets, VEC encompass a heterogeneous population of cells with different phenotypic characteristics and a propensity towards pro-calcific behavior. VEC activation and dysfunction have been linked with: (1) impaired nitric oxide (NO) generation; (2) increased oxidative stress; (3) enhanced pro-inflammatory pathways; (4) secretion of pro-osteogenic growth factors, and (5) EndMT [17]. Although compared to VIC the involvement of VEC in CAVD has been less convincingly verified and studied, recently it was found that PTH promotes osteoblastic differentiation of human umbilical vein endothelial cells via the extracellular signal-regulated protein kinase 1/2 (ERK1/2) and the nuclear factor-κB signaling pathways [18].

Thus, in the present study, we hypothesized that PTH could also have a role in valve calcification by inducing valvular endothelial cells dysfunction that, in turn, could affect valvular interstitial cells’ phenotype. We present here data showing that PTH produces VEC dysfunction that further stimulates VIC to differentiate into a pro-osteogenic phenotype, data that points to PTH as a novel target in the therapy of valvular diseases.

## 2. Results

### 2.1. Detection of Parathyroid Hormone Receptor Expression in Human Aortic Valve and Cultured Valvular Endothelial Cells

It is known that PTH acts by binding to the PTH receptor (PTH1R), and it can only affect cells that express this receptor. Therefore, before investigating the effect of PTH on VEC and its possible role in valve calcification, we first determined if PTH1R is present on the surface of both cultured human VEC and in human aortic valve tissue sections, with or without calcium deposits. We found that PTH1R was present on both aortic VEC (Figure 1A(a,c)) and valve sections (Figure 1B(a,c)); in Figure 1A(b,d), the Alexa-coupled secondary antibody staining is presented as a negative control. As shown, sections with extended calcium deposits exhibit increased expression of PTH1R around the calcium deposits (Figure 1B(a)) compared to areas without calcium deposits (Figure 1B(c)). The presence of calcium deposits on sections was demonstrated by Alizarin staining (Figure 1B(b,d)).

### 2.2. PTH Induces Dysfunction of Human Valvular Endothelial Cells

Valvular endothelium produces endothelium-derived relaxing factors and mediates valvular homeostasis [19]. VEC-derived nitric oxide is involved in paracrine signaling [20,21], and a decrease in NO bioavailability is associated with endothelial dysfunction.

To investigate if PTH affects VEC functions, we measured the NO production in cells exposed for 7 days to PTH. The results revealed that NO production was decreased in VEC exposed to PTH (Figure 2A). Moreover, gene expression of endothelial NO synthase (eNOS)—primarily responsible for the generation of NO in the vascular endothelium—is significantly reduced in PTH-exposed VEC (Figure 2B). Although NO has a central role in endothelial cells’ homeostasis, the mechanisms underlying impaired endothelial function in various disease states are likely multifactorial. Therefore, there is growing evidence showing that oxidative stress (defined as an imbalance between endogenous oxidants and antioxidants) contributes to mechanisms of vascular dysfunction [22]. Increased oxidative stress is due at least in part to a reduction in expression and activity of antioxidant enzymes and perhaps to uncoupled NOS activity. As expected, reactive oxygen species (ROS) level was increased in VEC treated with PTH (Figure 2C). Since catalase is one of the crucial antioxidant enzymes that mitigates oxidative stress (abolishing cellular hydrogen peroxide), we investigated the effect of PTH on catalase activity. We detected a significant reduction in catalase activity in VEC exposed to PTH (Figure 2D), suggesting a dysfunctional effect of PTH on VEC.

### 2.3. Energetic Metabolism in VEC Is Modified by PTH

A dysfunctional endothelial cell may also exhibit an altered metabolism. To determine the metabolic phenotypes of VEC under PTH exposure, we performed Seahorse bioenergetics measurements, using a mitochondrial stress test after 48 h or 7 days of cells exposed to PTH. This test measures oxygen consumption rate (OCR) at baseline levels after addition of (i) oligomycin A (coupled respiration), (ii) FCCP (maximum respiration), and (iii) respiratory inhibition with antimycin A and rotenone (non-mitochondrial oxygen consumption).

We found that basal respiration and OCR linked to ATP production were similar in control cells and cells exposed to PTH for 48 h (Figure 3A). However, after 7 days of exposure to PTH, a significant reduction in mitochondrial respiration (OCR) of PTH-exposed VEC versus control VEC was detected. This decrease was linked to both cellular ATP production (in response to oligomycin) and disruption of the mitochondrial membrane potential (in response to FCCP). However, since lactic acid accumulates in cells maintained in culture for a long time, the observed reduction in mitochondrial respiration may be a result of both cell culture and PTH treatment. Non-mitochondrial respiration driven by processes outside the mitochondria (rotenone/antimycin A addition) was also lower in VEC exposed to PTH (Figure 3C).

However, the switch from mitochondrial respiration to glycolysis was increased in PTH cells as compared to control cells. Uncoupling the mitochondrial respiratory chain and the oxidative phosphorylation with FCCP also induced a switch to glycolysis, with an increased glycolytic rate in PTH cells compared with control cells (Figure 3B,D). The increased glycolysis was accompanied by a reduction in glucose levels after 7 days, as compared with 48 h, in the conditioned media of VEC exposed to PTH (Appendix A). Together, these results showed that PTH induces a glycolytic phenotype and inhibits mitochondrial function in VEC.

### 2.4. PTH Does Not Modify the Expression of Endothelial and Mesenchymal Markers: CD31, vWF and α-SMA in VEC

Recent studies suggested that under certain conditions, VEC could participate in aortic valve calcification via EndMT [15]. To investigate if PTH may promote VEC transition to mesenchymal-like cells, we analyzed the expression of endothelial markers CD31 and vWF, and mesenchymal marker α-SMA in control VEC or VEC exposed for 7 days to PTH. Immunofluorescence results indicated that while α-SMA is rarely expressed by control VEC or VEC exposed to PTH, CD31 and vWF are abundantly expressed in cells in both conditions (Figure 4). We did not observe a significant modification of CD31 and vWF markers in endothelial cells treated with PTH as compared with control VEC (Figure 4).

### 2.5. PTH Activates p38MAPK and ERK1/2 Kinases and Upregulates Molecules Associated with Valve Calcification in VEC

Among other important roles, the MAPKs were revealed as key players in skeletal development and bone homeostasis, particularly affecting osteoblast commitment and differentiation [22]. Between the three classic MAPKs, scientific evidence found that activation of p38 and ERK1/2 led to the modulation of different osteogenic molecules [23]. Since our data indicate that PTH induces VEC dysfunctions, we searched for the signaling mechanism triggered by PTH in these cells. We found a significant increase in the phosphorylation form of both p38MAPK and ERK1/2, an indication that these kinases were activated by 7 days exposure of VEC to PTH (Figure 5A,B). The results were supported by experiments showing that treatment of VEC with PTH in the presence of SB239063—a specific inhibitor for p38MAPK—significantly decreased p38MAPK but not ERK1/2 activation (Figure 5A,B).

There is also evidence that PTH affects different cells by activating p38MAPK and that activation of p38 mediates PTH-induced stimulation of genes that play a critical role in matrix mineralization in osteoblastic cells [24]. We evaluated the effect of PTH on some of the molecules associated with valve calcification and the effect of p38MAPK inhibition. The qPCR data showed that osteogenic molecules morphogenetic protein-2 (BMP-2), osteopontin (OPN) and bone sialoprotein (BSP) expression (Figure 5C–E), but not transforming growth factor-β1 (TGF-β1) (Figure 5F), were significantly induced by PTH and that inhibition of p38MAPK using SB239063 significantly reduced OPN and BSP gene expression (Figure 5D,E). Moreover, both investigated transcription factors SOX9 and Runx-2 were also induced by PTH in VEC, dependent on p38MAPK (Figure 5G,H).

Protein expression of osteogenic molecules BMP-2, OPN, BSP and TFG-β (Figure 6A–D), as well as osteogenic transcription factor Runx-2 (Figure 6E), were also significantly increased in cells treated with PTH. Moreover, the inhibitor SB239063 significantly reduces the protein expression of the OPN and BSP osteogenic molecules (Figure 6B,C), and of RUNX-2 transcription factor (Figure 6E).

### 2.6. PTH Does Not Have an Angiogenic Effect on Endothelial Cells

Since angiogenesis is a hallmark of fibrocalcific aortic valve disease and previous studies found that PTH is involved in angiogenesis [25], we tested the angiogenic properties of the secretome-containing mediators released by PTH-exposed VEC. Results showed that PTH does not affect the production of pro-angiogenic factors of VEC. Thus, conditioned media (CM) from control or PTH-exposed cells induced similar tube-like structures by both HUVEC and VEC (Figure 7A,C). This data is also supported by the results showing that VEGF gene expression was not significantly affected in VEC exposed to PTH for 7 days (Figure 7B).

### 2.7. Mediators Secreted by Endothelial Cells Exposed to PTH Induce the Switch of VIC to an Osteogenic Phenotype

Calcification is associated with a decrease in valve collagen content [26]. Moreover, disruption of the endogenous collagen structure in aortic valves is sufficient to induce pathological consequences in valve leaflets [27], highlighting the importance of collagen in maintaining valve homeostasis. Moreover, we and others found that BMP-2/4 pathways are associated with an osteogenic phenotype of VIC [28,29]. To investigate if dysfunctional VEC cause the switch of VIC towards an osteogenic phenotype, we examined the effect of VEC secretome on cultured VIC. Results showed that the secretome of PTH-exposed VEC (VIC+mPTH) decreased protein expression of collagen I and collagen III (Figure 8A,B) and increased protein expression of both BMP-2 and BMP-4 (Figure 8C,D) in VIC. In addition, the levels of osteogenic molecules osteocalcin and TGF-β1 released by VIC were significantly increased (Figure 8E,F). CM from VEC or VEC+PTH or VIC exposed directly to PTH does not exhibit increased levels of osteocalcin or TGF-β1 (Appendix A), suggesting that factors produced by dysfunctional VEC stimulate VIC to produce these osteogenic factors.

## 3. Discussion

The increased incidence of valvular diseases and in particular of calcific aortic valve disease in the last 30–40 years [30] and the need for more data on the evolution, factors and relationships between the cells involved (VEC and VIC) prompted us to investigate whether PTH has a role in the interaction between these two cell types in disease initiation or progression. Although the recent studies suggest that PTH could directly induce endothelial dysfunction and contribute to vascular calcification [18,31,32], the effect of this hormone on human valvular endothelial cells and the effect of dysfunctional VEC on VIC osteogenic phenotype remain unclear.

In this study, our findings suggest that chronic exposure of VEC to PTH produces dysfunction, as reflected in: (i) a decline in the production of nitric oxide; (ii) impairment of anti-oxidative capacities; (iii) increased glycolysis capacity; (iv) increased expression of molecules associated with valve calcification and (v) induction of an osteogenic profile to VIC (Figure 9).

Nitric oxide is a multifunctional signaling molecule involved in the maintenance of metabolic and cardiovascular homeostasis. It is a potent endogenous vasodilator with an important role in suppressing the formation of vascular lesions [33] and in protecting VIC and the aortic valve from calcification [20]. Our results showed that PTH impaired NO production in VEC, along with reduction of the expression of eNOS, which is responsible for most of the vascular NO production. A reduction of NO bioavailability can also be due to the inactivation of NO by ROS, which we found to be increased by PTH in VEC, a result in line with previous data on bovine endothelial cells (EC) [32]. In calcified valves, ROS production that propagates intracellular signals and iron-catalyzed oxidative damage is increased in regions with calcium deposits and is accompanied by a reduction in defense mechanisms involved in ROS removal [34]. Reduced catalase expression was reported in both calcified and non-calcified portions of diseased valves as compared to normal valves, but the mechanisms of catalase deficiency remain to be determined [34]. We found that reduced production of NO by PTH was accompanied by increased ROS, decreased activity of anti-oxidative enzyme catalase, and altered cellular metabolism. Therefore, after 7 days, PTH-treated VEC exhibited reduced mitochondrial respiration and an increased glycolytic phenotype. Importantly, the switch of metabolic processes could precede functional changes and disease development. Thus, increased glycolysis and reduced mitochondrial respiratory capacity induced by disturbed flow in human aortic EC was suggested to precede vascular inflammation and thereby atherosclerosis [35]. Mitochondrial dysfunction is of particular interest because EC mitochondria are more likely to serve primarily as essential signaling organelles, and mitochondrial damage appears to be dependent on ROS and causes loss of bioenergetic control, leading to vascular dysfunction [36]. Under normal physiological conditions, the main energy source of EC comes from glycolysis. A cellular metabolism based on glycolysis is also present in highly proliferative cancer cells. While cancer cells use glycolysis, since they are often exposed to a hypoxic environment, EC mainly relies on anaerobic glycolytic metabolism for energy production, despite their direct exposure to high levels of oxygen in the blood, which would allow them to metabolize glucose via the oxidative pathway. Although quiescent EC display a high glycolytic rate, it is lower than in proliferating EC [37] or that in inflammatory activation of EC [38]. Whereas for EC using glycolysis might be a strategy to protect themselves against oxidative stress arising from oxidative metabolism and also to transfer the maximal amounts of oxygen to perivascular cells, cancer cells prefer to use aerobic glycolysis for different reasons, including that intermediates from glycolysis can be used to synthesize macromolecules, such as nucleic acids, lipids and proteins, which are required for cancer growth and proliferation [39]. Our data reveal that similar to inflammatory activation of EC, PTH also induces VEC activation associated with increased glycolysis.

Previously, PTH was found to be an important component of EndMT in CKD-induced cardiovascular disorders [40]. Therefore, PTH-induced EndMT via the miR-29a-5p/GSAP/Notch1 pathway contributed to valvular calcification in rats with CKD [37]. Elevated concentration of PTH (10^−7^ M) was reported to induce EndMT in aortic EC [31]. Our data using VEC and PTH at a concentration of 100 times lower than in the above studies revealed no induction of the EndMT process, suggesting that PTH induces VEC dysfunction but not their transformation towards a mesenchymal phenotype. This discrepancy in results could be due to the different concentrations of PTH employed or to the vascular bed analyzed (aortic vs. valvular endothelial cells).

Interestingly, the activation of VEC induced by PTH was linked with increased expression of genes involved in the valve calcification process. Thus, we have found that the expression of osteogenic molecules, BMP-2, osteopontin and BSP, as well as of the transcription factors associated with osteogenesis process, Runx-2 and Sox-9, were significantly increased in VEC exposed for 7 days to PTH.

Among the candidate mechanisms triggered by PTH able to induce endothelial dysfunction, we chose to investigate the MAPKs signaling pathways because of the previous evidence on the implication of these pathways in the process [41] and particularly in calcific aortic valve disease [42]. We found that PTH induces activation of both p38MAPK and ERK1/2 in VEC. Interestingly, p38MAPK inhibitor SB239063 reduced gene and protein expression of OPN, BSP and Runx-2, suggesting that MAPKs signaling pathways are involved in the modulation of osteogenic proteins by PTH in VEC. As is well known, osteogenic proteins are also induced by SMAD-dependent signaling triggered by TGF-β or BMPs. Although BMP-2 was significantly increased by PTH in VEC, SMAD1/5/8 and SMAD2/4 expression weren’t significantly affected (Appendix A). Before, p38MAPK signaling was clearly implicated in skeletal development and bone homeostasis and particularly in osteoblast commitment and differentiation [23]. ERK1/2 was found to be expressed in osteoblasts and induced Runx-2 phosphorylation and osteocalcin expression [43]. Importantly, PTH induced osteogenic gene expression dependent on p38MAPK, whereas p38 deletion led to reduced ossification [44].

During the calcification process, in addition to expressing osteogenic molecules, EC can also be involved in angiogenesis. As is well known, normal aortic valves are avascular structures. However, in calcified valves, angiogenesis appears near calcified nodules in areas infiltrated with inflammatory cells [45]. To have a comprehensive image of the effects of PTH on VEC, we tested if CM from VEC exposed to PTH promotes angiogenic activity. Tube formation by either HUVEC or VEC was similar in the presence of CM from control or PTH-treated cells, suggesting that PTH does not affect the angiogenic capacity of endothelial cells. In good agreement with these results, we found that PTH does not affect the endothelial expression of VEGF, a potent angiogenic factor.

Our second objective was to investigate the relevance of the modifications induced by PTH in VEC with implications in valve pathology, namely whether dysfunctional VEC trigger the switch of VIC to an osteogenic phenotype. Previous studies, including ours, have demonstrated that VEC dictate VIC behavior and that VEC are necessary to regulate and maintain a physiological VIC phenotype [20,21,28]. Hence, our question was about the effect of dysfunctional VEC on VIC phenotype. Experiments showed that factors secreted by dysfunctional VEC increased the propensity of VIC to switch to an osteogenic phenotype, characterized by decreased collagen I and III protein expression and increased BMP-2 and 4—strong inducers of osteogenic differentiation. BMPs signal by binding to Ser/Thr kinase receptors (BMP type I and type II receptors), leading to phosphorylation of intra-cellular SMAD1/5/8 that translocate into the nucleus, where they cooperate with other DNA-binding proteins to regulate BMP target gene transcription, including Runx-2 [46]. Possibly as a consequence of increased BMP-2/4 expression, we found increased levels of TGF-β1 and osteocalcin released by VIC exposed to mediators secreted by dysfunctional VEC.

As mentioned above, VIC exposed to the secretome of PTH-treated VEC exhibited a decreased expression of collagen I and III—the main component (~50%) of the normal valve extracellular matrix. The data is in line with recent findings showing that collagen disruption or deficiency is sufficient to induce VIC activation to a pathological phenotype [27]. Thus, it is not surprising that depletion of collagen from the valve leaflet led to an increased expression of osteogenic markers, including osteocalcin and BSP [27]. These deleterious effects of dysfunctional VEC on VIC could be due to decreased levels of NO in VEC treated with PTH, as well as to different soluble mediators released by EC, including BMP-2, that activate VIC. Previously it was found that VEC-derived NO might be the potential inhibitor of the early phases of valve calcification [47], and impairment of NO generation by VEC can induce osteogenic activation of VIC [21].

In conclusion, the novel data resulting from our study show that PTH interferes with heart valve calcification, inducing valvular endothelial cells dysfunction, which in turn triggers the transition of valvular interstitial cells to an osteogenic phenotype. This data recommends PTH and the ensuing hyperparathyroidism as a potential target for the therapy of heart valve calcification.

## 4. Materials and Methods

### 4.1. Cell Culture

Human valvular endothelial cells

Primary VEC were isolated from noncalcified cusps of human calcified aortic valves obtained from CAVD patients, as we recently described [28]. The investigation was carried out according to the principles outlined in the Declaration of Helsinki for experiments involving human samples [48]. Participants gave their written informed consent by signing the appropriate paperwork respecting their anonymity and privacy rights. The Ethics Committee of the Institute of Cellular Biology and Pathology “Nicolae Simionescu” has approved the study.

Isolated VEC were cultured in Endothelial Cell Growth Medium with 20% FBS (Gibco, Waltham, MA, USA) and 100 U/mL penicillin, 100 μg/mL streptomycin and 50 μg/mL neomycin (Merck, Darmstadt, Germany).

After the first passage, CD31-positive VEC were purified using magnetic beads conjugated to monoclonal anti-human CD31 antibody (Miltenyi Biotech, Bergisch Gladbach, Germany), and VEC phenotype was confirmed by evaluation of endothelial-specific markers CD31 and vWF [27].

Human valvular interstitial cells

VIC were purchased (P10462, Innoprot, Bizkaia, Spain) and were cultured in Petri dishes or flasks in DMEM with 10% FBS and antibiotics (100 U/mL penicillin, 100 μg/mL streptomycin and 50 μg/mL neomycin).

### 4.2. Experimental design

(a) Exposure of VEC to PTH

VEC were seeded in 6-well plates at a density of 150,000 cells/well and cultured in Cascade Biologics Medium 200 (Gibco, Waltham, MA, USA) with 10% FBS and antibiotics. On the second day after passage, the cells were incubated with 10^−9^ M PTH (Parthyroid Hormone Fragment 1-34, P3796, Merck, Darmstadt, Germany) for 48 h or 7 days. Every two days the medium was replaced with PTH-containing fresh culture medium. The 10^−9^ M concentration was chosen because it was previously reported to affect aortic endothelial cells [30]. Moreover, in our initial tests aiming to find the appropriate PTH concentration for VEC (since a variety of PTH concentrations are mentioned in the literature, depending on cell type), this concentration (10^−9^ M) significantly increased ROS (Appendix A).

(b) Exposure of VIC to VEC secretome

VIC were seeded in 6-well plates at a density of 150,000 cells/well and cultured in DMEM medium with 10% FBS with antibiotics.

The secretome (culture medium) was collected from cultured VEC (control) or PTH-exposed VEC, as described above. VIC were then incubated for 7 days with the VEC secretome and investigated for protein expression of osteogenic molecules using western blot or ELISA assays.

### 4.3. Immunofluorescence of Valve Sections and VEC

Valve tissue was obtained from surgically removed valves from patients with calcific aortic valve disease. Sections were washed with PBS and fixed in 4% PFA for 1 h at RT. Cryoprotection was achieved by successive incubation in PBS and 5, 10, 20 and 50% glycerol solutions. After 6 washes in 3% sucrose in 0.1 M phosphate buffer, the samples were immersed in OCT compound for 1 h (Neg-50, Thermo Scientific, Waltham, MA, USA) and 7 μm cryosections were cut using a Leica Cryotome and further processed. Some sections were used for immunofluorescence, while others were used for staining with Alizarin Red (Merck, Darmstadt, Germany, A5533-25G).

For immunofluorescence studies, cryosections of human valves or cultured VEC were fixed with 4% paraformaldehyde, permeabilized with PBS+0.5% Triton for 20′ in advance and blocked in TBS Blotto A (SC-2333, Santa Cruz, Dallas, TX, USA) with 3% BSA, 0.3% Triton X-100 and 1% cold fish gelatin (G7041, Merck, Darmstadt, Germany). Then, the sections/cells were incubated overnight at 4 °C with primary anti-human PTH1R (G7041, Merck, Darmstadt, Germany), anti-PECAM (SC376764, Santa Cruz, Dallas, Texas, USA), anti-α-SMA (SC32251, Santa Cruz, Dallas, TX, USA) and anti-von Willebrand Factor (PA5-16694, Invitrogen, Waltham, MA, USA) diluted in PBS with 0.3 Triton and 1% BSA. The next day, samples were washed and incubated with Chicken anti-Mouse Alexa Fluor 594 conjugated secondary antibody (A-21201, Thermo Scientific, Waltham, MA, USA) or Goat Anti-Rabbit FITC (ab6717, Abcam, Cambridge, UK) at a dilution of 1:1000 for 1 h 30′ at RT in the dark. Nuclei were stained with DAPI (D9542, Merck, Darmstadt, Germany) and samples were mounted in Fluoromount-G (Thermo Scientific, Waltham, MA, USA). Images were acquired using a fluorescence microscope (Olympus IX81, Shinjuku City, Tokyo, Japan) equipped with an XM10 camera. Images were processed using ImageJ software.

Calcium deposits from valvular sections were stained with Alizarin Red Solution 2%, pH 4.1–4.3 for 30 min at RT. After the staining step, slides were washed, dehydrated in acetone and mounted in mounting medium. Catalase activity was measured using BioVision Catalase Activity Colorimetric/Fluorometric Assay Kit (K773-100, BioVision, Milpitas, CA, USA) according to the manufacturer’s instructions for cell lysate. Catalase activity was normalized to the protein quantity used in the assay and was expressed as nmol/μg protein.

### 4.4. Nitic Oxide (NO) Assays

NO concentration was determined in cell lysate using Nitric Oxide Colorimetric Assay Kit (E-BC-K035-M, Elabscience, Wuhan, China) according to the manufacturer’s instructions. NO levels were normalized to protein and expressed as nmol/g protein.

### 4.5. Evaluation of ROS Production

VEC were cultured at a density of 5 × 10^4^ per well in 24-well plates. After incubation with PTH, cells were incubated with 5 μM 2′,7′-Dichlorofluorescin diacetate (DCFDA, Merck, Darmstadt, Germany) for 30 min at 37 °C for total ROS detection. Next, cells were washed twice with PBS and fresh medium was added. Fluorescence was immediately measured by a plate reader (Tecan Infinite 200 Pro, Männedorf, Switzerland) using excitation/emission wavelengths of 492/520 nm.

### 4.6. Cell Energy Assay

A Seahorse XFp Extracellular Flux Analyzer (Agilent Technologies, Santa Clara, CA, USA) was used to measure the oxygen consumption rate (OCR) and extracellular acidification rate (ECAR) in control VEC or VEC exposed to PTH for 48 h or 7 days. VEC were added at 7 × 10^3^ cells/well onto a XFp plate and stimulated for 48 h or 7 days with PTH at a concentration of 10^−9^ M. Then, the medium was changed over to Seahorse XF DMEM base medium, without phenol red (cat # 103335-100, Agilent Technologies, Santa Clara, CA, USA), smented with 10 mM Glucose, 2 mM Glutamine and 1 mM Sodium Pyruvate pH 7.4, and metabolic parameters were measured using the Mitochondrial Stress kit (Cat # 103015-100, Agilent Technologies, Santa Clara, CA, USA), according to manufacturer’s protocol. Oligomycin (ATP Synthase inhibitor, 1.0 mM), mitochondrial uncoupler carbonyl cyanide 4-(trifluoromethoxy) phenylhydrazone (FCCP, mitochondrial membrane depolarizer, 1.5 mM) and a mix of 0.5 mM of each rotenone (complex I inhibitor) and antimycin A (complex III inhibitor) were added in the ports of Seahorse flux plate. Data presented are representative of 3 independent experiments with 3 replicate/experimental conditions within each experiment.

### 4.7. Quantitative RT-PCR

Gene expression of BMP-2, OPN, BSP, TGF-β1, SRY-Box Transcription Factor 9 (Sox9), Runt-Related Transcription Factor 2 (Runx-2), vascular endothelial growth factor (VEGF) and collagen I/III was evaluated using qPCR. For each experiment, mRNA was extracted from control or PTH treated-cells, using PureLink RNA mini-Kit (Ambion, Carlsbad, CA, USA) or Trizol (Merck, Darmstadt, Germany). First-strand cDNA synthesis was performed employing 1 μg of total RNA and MMLV reverse transcriptase (Invitrogen, Walthan, MA, USA). Assessment of gene expression was done by amplification of cDNA using a LightCycler 480 RT-PCR System (Roche, Basel, Switzerland) and SYBR Green I chemistry. Primer sequences are shown in Table 1. Relative quantification was done by comparative CT method and expressed as arbitrary units. β2-microglobulin was used as a reporter gene for all the investigated molecules.

### 4.8. Western Blot

Expression of collagen I/III, osteogenic and signaling molecules was assessed in the lysate of VEC or VIC, homogenized in Laemmli electrophoresis buffer. Cell lysates were subjected to electrophoresis and immunoblot assay as previously described [27], using specific antibodies: pERK1/2 (197G2), pp38MAPK (GTX133460), BMP-2/-4 (MAB 3552), TGF-β1 (MAB-240), OPN (MAB14331), BSP (AF-4014), Runx-2 (PA1-41519), collagen I (PA1-85319)/III (PA5-27828) and GAPDH (AM 4134). Signals were visualized using SuperSignal West Dura (34076, Thermo Scientific, Waltham, MA, USA). Imaging of protein bands was obtained using a digital detection system (ImageQuant LAS 4000, Fujifilm, Tokyo, Japan) and quantified by densitometry, with Image J software.

### 4.9. Matrigel Assay

The formation of tube-like structures by human vascular endothelial cells (HUVEC—EA926 line) or VEC was assessed employing a solid gel of basement membrane proteins and growth factors (In Vitro Angiogenesis Assay Kit, ECM 625 Merck, Darmstadt, Germany). The bottoms of 96-well culture plates were coated with Matrigel (50 µL per well) and incubated for 1 h at 37 °C for polymerization. The gels were overlaid with 100 µL of conditioned media (CM) collected from control or PTH-exposed VEC for 7 days together with 20,000 endothelial cells/well.

Tube formation was monitored by an inverted phase contrast microscope (Olympus IX81, Shinjuku City, Tokyo, Japan) equipped with an XM10 camera. Tubule branching points, meshes and the total length were analyzed using the Angiogenesis Analyzer plugin of Image J program.

### 4.10. Enzyme-Linked Immunosorbent Assay (ELISA)

Soluble osteocalcin or TGF-β1 levels were quantified in conditioned media of VIC collected after exposure to VEC secretome (as described in the Experimental design section), using ELISA DuoSet kits (DY240-05 and DY1419-05 R&D systems, Abingdon, UK), following manufacturer’s instructions.

### 4.11. Statistical Analyses

All statistical analysis was performed with GraphPad Prism 7.0 software (San Diego, CA, USA) with data points denoting mean ± standard deviation (SD). Statistical significance is shown as p-values obtained via a two-tailed Student’s *t*-test when comparing two experimental groups and analysis of variance (One-Way ANOVA) with multiple comparisons when comparing more than two groups. A *p*-value of *p* < 0.05 was considered statistically significant.

## Figures and Tables

**Figure 1 ijms-23-03776-f001:**
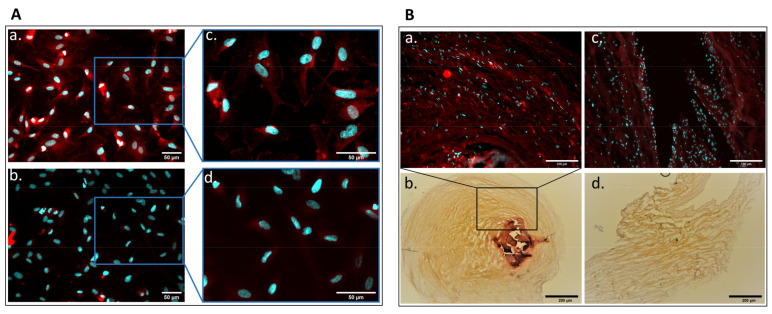
Parathyroid hormone receptor (PTHR) is present on cultured valvular endothelial cells and human heart valve. The expression of PTHR was evaluated by immunofluorescence using a specific primary PTHR antibody and Alexa-coupled secondary antibody (red) on cultured VEC (**A**(**a**) and inset in **A**(**c**)) a human valve section (**B**) with calcium deposits (**B**(**a**)) or without calcium deposits (**B**(**c**)). The negative control, Alexa-coupled secondary antibody staining, is present in Ab and inset in Ad. Nuclei were stained with DAPI (blue staining). The calcific zones of the valve section are marked by Alizarin Red staining (**B**(**b**,**d**)).

**Figure 2 ijms-23-03776-f002:**
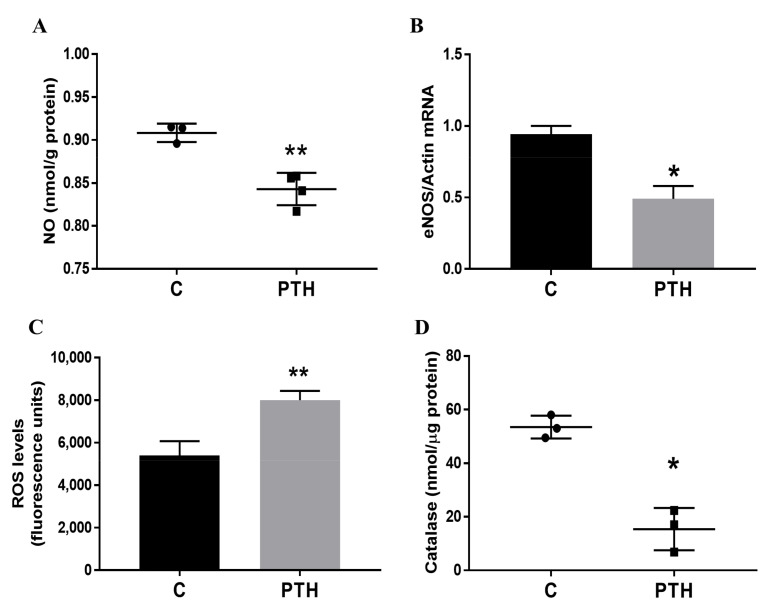
PTH induces VEC dysfunction, altering NO and ROS production, eNOS expression and catalase activity. (**A**) NO levels in VEC following exposure to PTH (10^−9^ M) for 7 days. (**B**) Gene expression of eNOS in control or PTH-exposed VEC. (**C**) Evaluation of ROS production in VEC, in response to PTH. The total amount of ROS was determined by incubating the cells with the fluorescent probe DCFDA. The ROS levels are expressed as relative fluorescence; (**D**) Catalase activity was determined at 570 nm and the values are expressed as nmol. Three independent experiments were performed * *p* < 0.05 and ** *p* < 0.01, PTH vs. the untreated control VEC.

**Figure 3 ijms-23-03776-f003:**
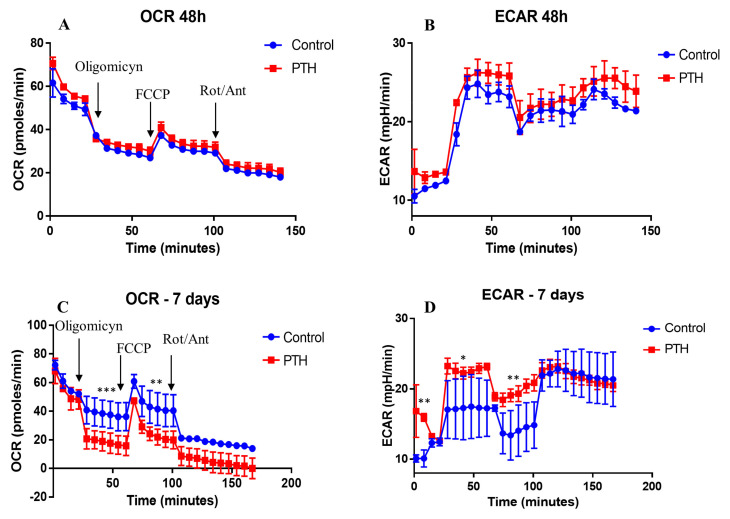
The effect of PTH on mitochondrial respiration and glycolysis in VEC. (**A**,**C**) Mitochondrial respiration presented as OCR (a measure of oxygen utilization in cells) at 48 h and 7 days after VEC exposure to PTH. Arrows indicate injection of oligomycin (0.5 µM), carbonyl cyanide-4-(trifluoromethoxy) phenylhydrazone (FCCP−1 μM) and rotenone/antimycin (1 μM). Each data point represents an OCR measurement. (**B**,**D**) ECAR—a measure of lactic acid levels, generated by anaerobic glycolysis, quantified under basal conditions and as a response to the same inhibitors, oligomycin, FCCP and rotenone +antimycin. Data are expressed as means ± SEM, *n* = 4 independent experiments, * *p* < 0.05, ** *p* < 0.01, *** *p* < 0.001 control vs. PTH-exposed VEC.

**Figure 4 ijms-23-03776-f004:**
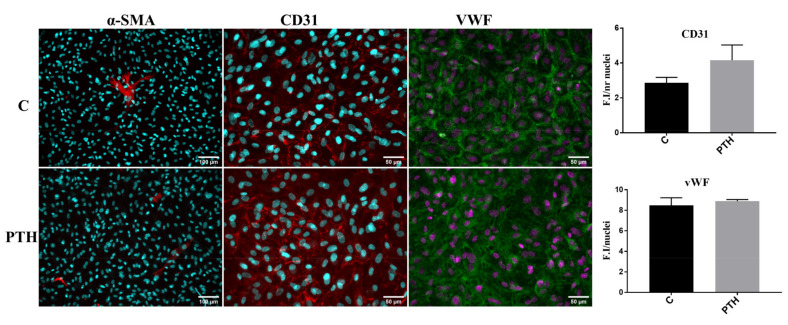
Expression of markers associated with EndMT transition of endothelial cells. Immunofluorescence staining of mesenchymal marker—αSMA (red) and of endothelial markers—CD31 (red) and vWF (green) in valvular endothelial cells control (C) or after 7 days of PTH exposure (PTH). Mean fluorescence intensity (FI) of images was calculated for the fluorophore and normalized to the number of cell nuclei stained with DAPI (cyan or purple staining).

**Figure 5 ijms-23-03776-f005:**
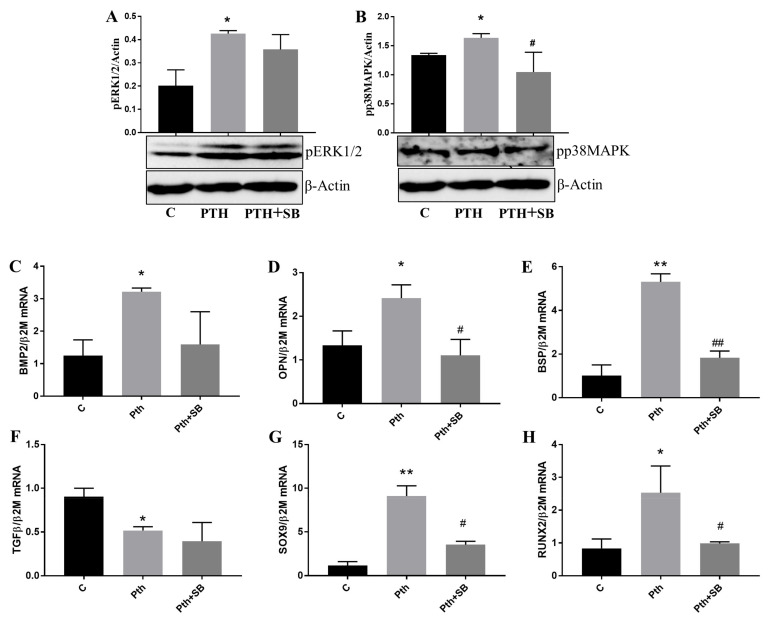
PTH activates MAPK signaling pathways and induces gene expression of osteogenic molecules in VEC. (**A**,**B**) The effect of PTH on the expression of phosphorylated ERK1/2 (**A**) and p38MAPK (**B**) in valvular endothelial cells. Cell lysates from control—C or PTH-treated VEC -PTH, cultured for 7 days in the presence or absence of SB-239063, were analyzed by western blot. (**C**–**H**) Gene expression of osteogenic molecules BMP2 (**C**), OPN (**D**), BSP (**E**), TGF-β1 (**F**), SOX-9 (**G**) and RUNX-2 (**H**) expressed by human VEC exposed to PTH in the presence or absence of p38MAPK inhibitor SB239063 for 7 days, as evaluated by qPCR. The mRNA of osteogenic molecules was normalized to β2 -Microglobulin (β2M) mRNA. *n* = 3, * *p* < 0.05, ** *p* < 0.01 (PTH vs. C); **^#^**
*p* < 0.05, **^##^**
*p* < 0.01 (PTH+SB vs. PTH).

**Figure 6 ijms-23-03776-f006:**
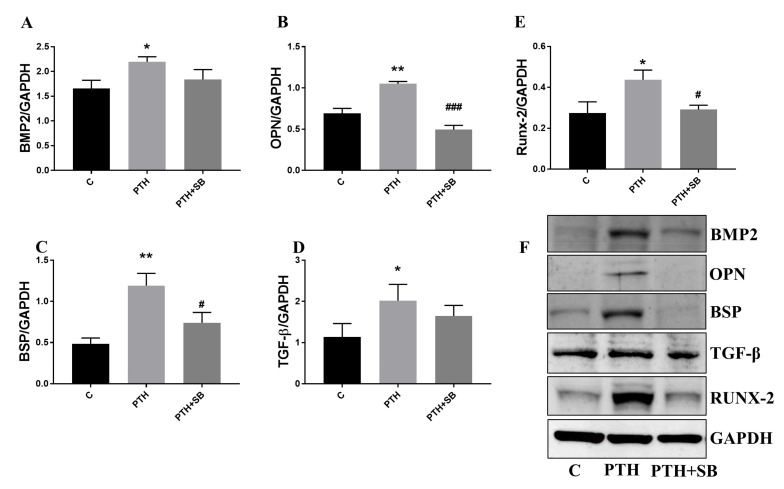
Protein expression of osteogenic molecules expressed by VEC exposed to PTH. (**A**–**E**) Quantification of protein expression of BMP-2, OPN, BSP, TGF-β1 and Runx-2 in VEC as determined by western blot. (**F**) Representative western blot images for investigated molecules are presented. *n* = 3, * *p* < 0.05, ** *p* < 0.01 (control VEC vs. VEC exposed to PTH for 7 days); **^#^**
*p* < 0.05, **^###^**
*p* < 0.001 (PTH+SB vs. PTH).

**Figure 7 ijms-23-03776-f007:**
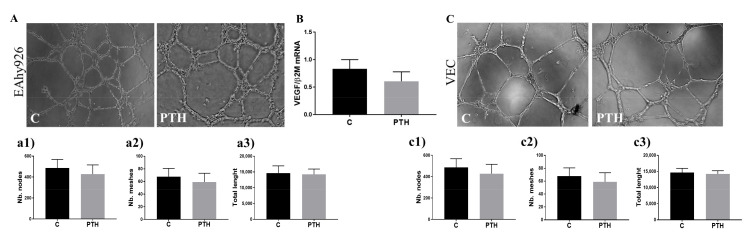
Effect of conditioned media released by control and PTH-exposed VEC on endothelial cells tube formation. (**A**). Tube-like formation by umbilical endothelial cells (EAhy926) induced by CM of control VEC or VEC exposed to PTH. Number of nodes, meshes and total length of tube-like structures are quantified (**a1**–**a3**). (**B**) VEGF gene expression in control or PTH exposed VEC. (**C**). Tube formation by VEC in the presence of CM from control VEC or VEC exposed to PTH. The number of nodes, meshes and total length of tube-like structures are quantified (**c1**–**c3**). *n* = 3.

**Figure 8 ijms-23-03776-f008:**
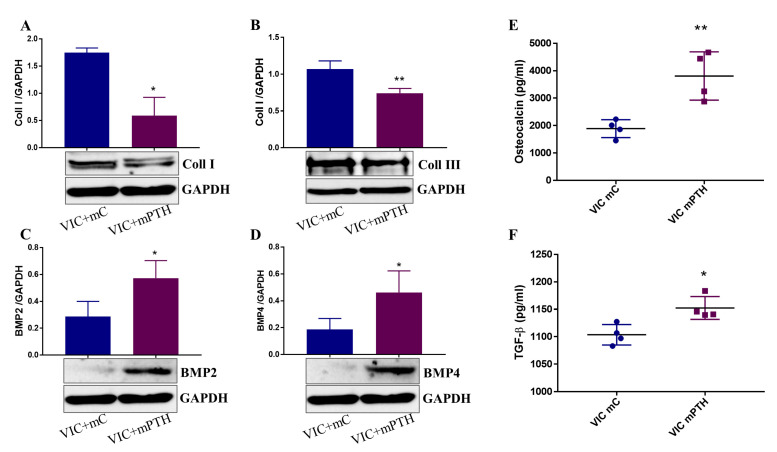
Conditioned media from VEC affects VIC phenotype. (**A**–**D**) Protein expression of collagen I/III and of BMP-2/-4 expressed by VIC exposed to CM from control VEC (VIC+mC) or to CM from VEC exposed to PTH (VIC+mPTH), as determined by western blot assay. Representative western blot images are shown under the graphs. *n* = 3, * *p* < 0.05, ** *p* < 0.01. (**E**,**F**). Quantification (ELISA assay) of osteocalcin and TGF-β1 released by VIC exposed to CM from VEC. *n* = 4, * *p* < 0.05, ** *p* < 0.01.

**Figure 9 ijms-23-03776-f009:**
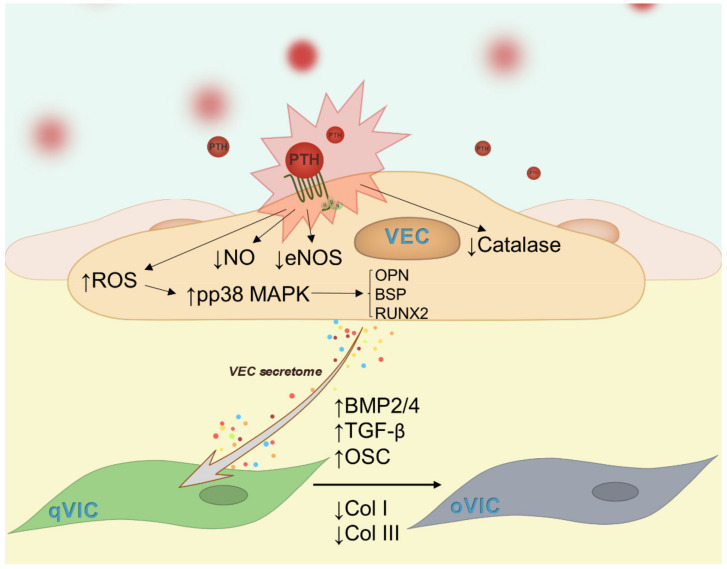
Summary figure. Exposure of VEC to PTH induces dysfunction to VEC, as revealed by decreased levels of NO, eNOS and catalase activity, increased ROS, activation of p38MAPK and overexpression of osteogenic molecules: osteopontin, BSP and Runx2. Factors released by dysfunctional VEC further stimulate VIC to differentiate into a pro-osteogenic phenotype, with increased expression of BMP-2/4, TGF-β1 and osteocalcin and decreased collagen I and III, modifications associated with the calcification process.

**Table 1 ijms-23-03776-t001:** The sequences of oligonucleotide primers used for evaluation of gene expression.

Gene	GenBank^®^ Accession Number	Sequences of Oligonucleotide Primers	Predicted Size (bp)
TGF-β 1	NM_000660.7	Fw: 5′-ccacctgcaagaccatcgac-3′ Rv: 3′-ctggcgagccttagtttggac-5′	91
Osteopontin	NM_001040058.2	Fw: 5′-gaagtttcgcagacctgacat-3′ Rv: 5′-gtatgcaccattcaactcctcg-3′	91
BMP-2	NM_001200.4	Fw: 5′-actaccagaaacgagtgggaa-3′ Rv: 5′-gcatctgttctcggaaaacct-3′	113
BSP	NM_004967.4	Fw:5′gaacctcgtggggacaattac-3′ Rv:3′ catcatagccatcgtagccttg-5′	79
VEGF	NM_001025366.3	Fw:5′-agggcagaatcatcacgaagt-3′ Rv: 3′agggtctcgattggatggca-5′	75
Sox9	NM_000346.4	Fw:5′agcgaacgcacatcaagac-3′ Rv:3′ ctgtaggcgatctgttgggg-5′	85
RUNX2	NM_001024630.4	Fw: 5′-ccgcctcagtgatttagggc-3′ Rv: 5′-gggtctgtaatctgactctgtcc-3′	132
β2-microglobulin	NM_001101.4	Fw: 5′-catgtacgttgctatccaggc-3′ Rv: 5′-ctccttaatgtcacgcacgat-3′	250

## Data Availability

The data presented in this study are available on request from the corresponding author.

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
