# Peer review of "Parathyroid Hormone Induces Human Valvular Endothelial Cells Dysfunction That Impacts the Osteogenic Phenotype of Valvular Interstitial Cells"

_ijms, 2022, doi:10.3390/ijms23073776_

Round 1

Reviewer 1 Report

In this manuscript, Vadana et al. investigated the effects of PTH on human valvular endothelial cells (VEC) and whether such effects would further lead to the valvular interstitial cells (VIC) phenotype alteration. The study is of interest; however, a few comments to be addressed before the publication.

The abstract needs to be revised, which should be concise.

The authors have shown the arathyroid hormone receptor (PTHR) present on cultured valvular endothelial cells and human heart valve. A straightforward mechanism exploration should be the gene silencing of PTHR in VEC using for instance shRNA plasmids.

The authors have evaluated different endpoints to verify VEC dysfunction induced by PTH. It is encouraged to draw a schematic figure to illustrate all the evaluated markers, including NO, eNOS, ROS, mitochondrial dysfunction, kinase pathways, and the correlations among them. The exposure scenario of VIC (i.e. VEC secretome) can be included as well to better guide the readers.

The PTH-treated VEC exhibited a reduced mitochondrial respiration and an increased glycolytic phenotype. This reprogramming of cellular metabolism is also a hallmark of cancerous cells. Could the authors comment on this?

Reviewer 2 Report

This is an interesting and well-written study that investigated whether parathyroid hormone (PTH) induces human valvular endothelial cells (VEC) dysfunction and the effects on reprogramming and switching valvular interstitial cells (VIC) myofibroblastic to osteogenic phenotype using in vitro approach. The study is an asset to understanding the molecular mechanisms regulating VIC pathological remodeling, calcification as well as cellular communication between VEC and VIC. Vadana and colleagues exposed VIC to the secretome of PTH-treated VEC followed by molecular analysis. The results indicate that parathyroid hormone receptor is expressed in aortic VEC. The authors state that expression of PTH receptor was higher in areas of extended calcium deposition. Interesting results also indicate that PTH reduced endothelial NO synthase (eNOS) expression, nitric oxide (NO) production and reduced catalase enzyme activity as well as increased reactive oxygen species (ROS) production in VEC. The results indicate that PTH regulates VEC cellular metabolism by reducing mitochondrial functions in VEC, mitochondrial oxygen consumption and increasing glycolytic rate and lactic acid production. The results indicate that PTH upregulated osteogenic differentiation programming and while PTH did not affect pro-angiogenic factors in VEC, secretome of PTH-treated VEC downregulated collagen I and collagen III and upregulated osteogenic factors BMP-2 and BMP-4 in VIC. The authors did an outstanding work with experimental evidence supporting their conclusions. However, there are some concerns that need to be addressed.

  • Figure 1 should be reorganized as follow:
  • PTH receptor expression in VEC.
  • PTH receptor expression in human valve section without calcium.
  • PTH receptor expression in human valve section with calcium.
  • Negative control

  • The current images in the sections (Figure 1) are of variable magnifications, please put all images with same magnification, put scale bar on all images. In addition, make an inset of high magnification 80x or more for cells in b, c to show subcellular localization of PTH1R, whether it is localized at plasma membrane or in the nucleus.

  • It is difficult to make a solid conclusion from the images in figure 1 that at areas of extended calcium deposition, there was high expression of PTH1R. This should be corroborated with quantification from a number of cells from the two groups, with image J for example. This should include quantification of Alizarin red as well as PTH1R expression from serial sections.

  • Since figures 1E and 1F are not serial sections for 1C and 1D, the authors are therefore advised to show colocalization of calcium deposits and PTH receptor in the same section or serial sections that are right after each other (separated by 5 um).

  • Results section 2.3

Please add rationale for investigating the effects of PTH on mitochondrial dynamics and glycolysis. It is known that PTH increases aerobic glycolysis and lactate production in the presence of oxygen and decreases glucose entry into the tricarboxylic acid cycle (Warburg effect). Since the results in VEC also indicate reduction in mitochondrial oxygen consumption, increased glycolytic rate and lactic acid production, this part will be greatly improved by confirming these results by measuring glucose levels in culture media after at different times and doses of PTH treatment.

The study indicates that after 7 days of PTH exposure, reduction in mitochondrial respiration is observed. The same is true for untreated cells, as you can see from the data, that there is a trend for cells in culture to accumulate lactic acid altogether with reduced oxygen consumption after long time in culture indicating that this may not be a specific phenotype due to PTH treatment. Please explain this phenomenon.

The study could benefit from investigating the molecular basis of these metabolic changes such as investigating expression levels of hexokinase II, lactate dehydrogenase A, and pyruvate dehydrogenase kinase I.

Is there statistical difference for the different time points in 3A-3D. I suggest rephrasing text according to findings of statistical analysis.

  • Results part 2.5

Is the observed elevation in pERK1/2 and p38 primarily due to PTH or secondary to activation of FGF signaling?

It is established that FGF signaling, MAPK (pERK, p38) and RUNX2 regulate osteogenesis and osteoprogenitor cell proliferation. What is the effect of PTH on FGF10 and its receptor FGFR1b in VEC.

Assay for the readouts of BMP signaling as well as TGF signaling pathways namely pSmad1/5/8 and pSmad2/3 respectively is needed.

Saura and colleagues (2005, Circulation research) reported increased basal TGFβ expression and SMAD2 phosphorylation in eNOS knockout mice. The results in the current manuscript indicate that while eNOS and NO production were significantly reduced, TGFβ is also downregulated. Please clarify this discrepancy.

  • Discussion

The discission is lengthy and needs to be more sententious. In addition, please discuss the caveats of the current study. Use consistent font size and type.

  • Other comments

Line 49: change “present” to “expressed”, here and throughout the manuscript.

Line 93: regarding nomenclature of PTH receptor (PTHR), it should be standardized to PTH receptor type 1 here and throughout the manuscript.

Line 108, add reference for “Valvular endothelium produces endothelium-derived relaxing factors and mediates valvular homeostasis”

Line 173, 174: expressed in

Line 182, change violet to purple

Line 202: Specify TGF-β

Line 252, change “produce” to “cause”

Line 254, what is “mPTH”?

English articles “the” , “a” are not used correctly in the manuscript. Please review English.

Round 2

Reviewer 1 Report

The authors have done a great job to address all my questions. I recommend for publication in its current form.